# Uncovering spin-orbit coupling-independent hidden spin polarization of energy bands in antiferromagnets

Lin-Ding Yuan[1], Xiuwen Zhang [1], Carlos Mera Acosta[2] & Alex Zunger[1] ✉

Many textbook physical effects in crystals are enabled by some specific symmetries. In contrast to such 'apparent effects', 'hidden effect X' refers to the general condition where the nominal global system symmetry would disallow the effect X, whereas the symmetry of local sectors within the crystal would enable effect X. Known examples include the hidden Rashba and/or hidden Dresselhaus spin polarization that require spin-orbit coupling, but unlike their apparent counterparts are demonstrated to exist in non-magnetic systems even in inversion-symmetric crystals. Here, we discuss hidden spin polarization effect in collinear antiferromagnets without the requirement for spin-orbit coupling (SOC). Symmetry analysis suggests that antiferromagnets hosting such effect can be classified into six types depending on the global vs local symmetry. We identify which of the possible collinear antiferromagnetic compounds will harbor such hidden polarization and validate these symmetry enabling predictions with first-principles density functional calculations for several representative compounds. This will boost the theoretical and experimental efforts in finding new spin-polarized materials.

Many traditional textbook physical effects in crystals are enabled by some specific symmetries, encoded in the crystal space group. Such are the symmetry conditions for the apparent electric polarization which defines various order parameters such as in ferroelectricity[1], circular dichroism[2], and pyroelectricity[3]. Another example of effects enabled by the recognized global system symmetry is the removal of spin degeneracy of energy bands due to spin-orbit coupling (SOC) in non-magnetic crystals having broken inversion symmetry (such as the Rashba (R-1)[4] and Dresselhaus (D-1)[5] effects). When an effect is observed despite the needed enabling symmetry being absent, it is often assumed that the system contains some symmetry-altering imperfections.

In contrast, the "Hidden Effect X" in materials that are not supported by the nominal enabling symmetry, yet effect X exists locally. The "Hidden effect X" reflects the intrinsic properties of the perfect crystal rather than imperfections that would disappear when the crystal becomes perfect. The understanding of such hidden intrinsic effects is important as it can demystify peculiar observations of phenomena that are unexpected to exist based on the global symmetry of the system.

Examples of "Hidden Effect X" that is SOC-induced include (i) Rashba or Dresselhaus spin polarization, expected exclusively to occur in non-centrosymmetric crystals, but predicted[6,7] and observed[8-18] in centrosymmetric nonmagnetic crystals (denoted R-2 and D-2, respectively). Similar form of Hidden effect X are (ii) X = "anisotropic optical circular polarized luminescence" expected only in odd-layered transition-metal dichalcogenides but observed[19] also in even-layered crystals. Such effects were originally dismissed as being due to some extrinsic sample imperfection[20-22] but later on were shown to be an intrinsic property pertained to the individual layer[23]. (iii) X = "spin polarization" induced by SOC in antiferromagnetic systems. The effect is again expected only in non-centrosymmetric crystals (such as $BiCoO_3$[24]) but shown in centrosymmetric crystals (such as CuMnAs and $Mn_2Au$[25-28]) where combined symmetry of inversion and time reversal disallows splitting. Here, "centrosymmetric" means the crystal in the

[1]Renewable and Sustainable Energy Institute, University of Colorado, Boulder, CO 80309, USA. [2]Center for Natural and Human Sciences, Federal University of ABC, Santo Andre, São Paulo, Brazil. ✉e-mail: Alex.Zunger@colorado.edu

**a**

Hidden spin polarization in AFM

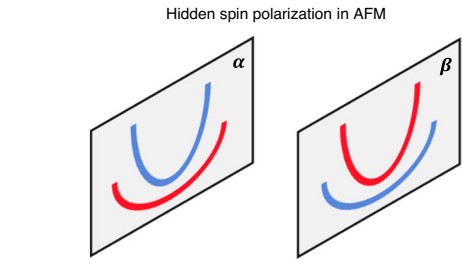

**b**　　Symmetry classification of the spin degenerate bulk

| Symmetry | SS w/o SOC | Magnetism | Prototype |
|---|---|---|---|
| $\Theta IT$(✓) and $UT$(✗) | No | AFM | SST-1 |
| $\Theta IT$(✓) and $UT$(✓) | No | AFM | SST-2 |
| $\Theta IT$(✗) and $UT$(✓) | No | AFM | SST-3 |

**c**　　Symmetry classification of spin split sector

| Symmetry | SS w/o SOC | Magnetism | Prototype |
|---|---|---|---|
| $\Theta IT$(✗) and $UT$(✗) | Yes | AFM | SST-4 |
| $\Theta IT$(✗) and $UT$(✗) | Yes | FM | SST-5 |

**Fig. 1 | Hidden spin polarization in collinear antiferromagnets without SOC.**
**a** SOC-independent hidden spin polarization schematically illustrated as two copies of spin split energy bands localized on sector-$\alpha$ and sector-$\beta$ but globally mutually compensate; (**b**) three prototypes of spin degenerate bulk; (**c**) two prototypes of spin split sector. Sectors in (**a**) are represented by color-shaded planes, the red and blue lines in the plane represent the spin-up and spin-down bands. The spin-splitting prototypes in (**b**) defined for bulk[39] is generalized for sectors in (**c**). Checkmark and cross in parentheses in (**b**) and (**c**) are used to indicate the presence or absence of the symmetry.

non-magnetic state has an inversion. Prominently, the hidden spin polarization in these compounds facilitates the electrical reversal of their antiferromagnetic ordering[25,29]. (iv) X = "anomalous Hall effect" induced by SOC expected only in odd-layered ferromagnetic $MnBi_2Te_4$ systems but observed in even-layered antiferromagnetic $MnBi_2Te_4$[30] systems via a perturbative applied electric field.

Here, we discuss a different form of hidden spin polarization effect (see Fig. 1a) whose corresponding apparent effect is independent of SOC[31–40]; And the hidden effect exists in antiferromagnetic materials where spin-up and spin-down bands are paired. This represents a step further beyond the already known hidden Rashba and hidden Dresselhaus spin-polarization that unavoidably require a sizable contribution from SOC. A careful analysis of the "global (bulk) vs local (sector)" symmetries suggests that antiferromagnets hosting the SOC-independent "hidden" spin polarization effect can be delineated into six types. We scrutinize a vast database of known collinear AFM materials and performed first-principles calculations on several selected candidate compounds assuming zero SOC. We show that such hidden, SOC-independent effects reflect the intrinsic properties of the perfect crystal rather than an effect due to imperfections. The interest in this SOC-independent hidden spin polarization effect stems both from the evolving of the fundamental understanding of general hidden effects in solids, and from the ability to extend the pool of useful materials for potential spintronic applications.

## Results

### Enabling symmetry conditions for SOC-independent apparent spin polarization in antiferromagnets

Symmetry is essential to understand the energy bands' degeneracy of a material. The symmetry conditions for apparent spin splitting or spin

polarization was pointed out recently in ref. [38]. This involved utilizing first a few individual symmetry operations: $U$ being a spin rotation of the SU(2) group acting on the spin 1/2 space that reverses the spin; $T$ being spatial translation; $\Theta$ being time reversal, and I being the spatial inversion. These individual operations are then used for constructing two symmetry products: a SOC-free magnetic symmetry $\Theta IT$, and a spin symmetry $UT$ (where the former product can be simplified to $\Theta I$ by proper choice of inversion center). SOC-independent spin splitting[38,39] would occur only when both symmetry products are simultaneously violated. Antiferromagnets with $\Theta IT$ symmetry[28,41,42] will not show such spin splitting. Such symmetry conditions disentangle the SOC-independent splitting from the SOC-induced splitting by considering the symmetry at the zero SOC limit[43–45], where spin and space are fully decoupled.

Given the symmetry conditions, it is thus possible to classify all different spin splitting prototypes[38,39] for magnetic materials. There are three prototypes with no apparent spin splitting effect: (1) AFM compounds that violate $UT$ but preserve $\Theta IT$ symmetry referred to as spin splitting prototype 1 (SST-1) antiferromagnets; (2) AFM compounds that preserve both $UT$ and $\Theta IT$ symmetry referred to as SST-2 antiferromagnets; (3) AFM compounds that preserve $UT$ but violate $\Theta IT$ symmetry referred to as SST-3 antiferromagnets. In addition, there are two prototypes with apparent spin splitting effects: (4) AFM compounds that violate both $UT$ and $\Theta IT$ symmetry referred to as SST-4 antiferromagnets; (5) Ferromagnetic (FM) compounds that violate both $UT$ and $\Theta IT$ symmetry referred to as SST-5 ferromagnets. The classification defined in bulk crystals[38,39] can be generalized to sectors of a bulk based on the local sector symmetry. Figure 1b,c summarizes the classification of "spin degenerate bulk" vs "spin-split sector". This will later be applied to describe the symmetry conditions and to define the different prototypes for the hidden spin polarization effect in antiferromagnets.

### Enabling symmetry conditions for hidden SOC-independent spin polarization in antiferromagnets

"Hidden spin polarization" is expected in collinear antiferromagnets when the bulk has zero net spin polarization, but its constituent sectors allow locally a spin splitting and spin polarization effect. Consider the combination of two possible prototypes constituting sector that gives hidden spin polarization locally but lead to three possible prototypes of the bulk symmetry (preserving either $\Theta IT$ or $UT$ or both) that disallows apparent spin polarization, one can then classify six hidden spin polarization cases. Following the previous classification of spin splitting prototypes for apparent spin degeneracy and apparent spin splitting[38,39], collinear antiferromagnetic materials with "hidden spin polarization" are those antiferromagnets whose bulk prototype being SST-I (I = 1, 2, 3) and constitute sector prototype being SST-J (J = 4, 5). Detailed discussions of the symmetry conditions for hidden spin polarization in collinear AFM are given in Supplementary Information Section A.

Figure 2 summarizes the six possible types of hidden spin polarization without SOC in antiferromagnets that are spin degenerate but contain spin split sectors (represented by color-shaped plane). Figure 2a–c illustrates the three cases where the spin degenerate antiferromagnets of SST-I (I = 1,2,3) can be decomposed into alternating ferromagnetic local sectors that locally violate both $UT$ and $\Theta IT$, thus allows spin splitting without SOC. FM materials that satisfy the conditions of violating both $UT$ and $\Theta IT$ (always true) are denoted as SST-5 in Fig. 1. The three magnetic-induced hidden spin polarization cases can then be denoted as (a) bulk SST-1 sector SST-5; (b) bulk SST-2 sector SST-5, and (c) bulk SST-3 sector SST-5. Figure 2d–f illustrates the three cases where the spin degenerate AFM of SST-I (I = 1, 2, 3) can be decomposed into alternating antiferromagnetic local sectors that locally violate both $UT$ and $\Theta IT$, thus allows spin splitting without SOC. AFM materials that satisfy the condition are denoted as SST-4 in Fig. 1.

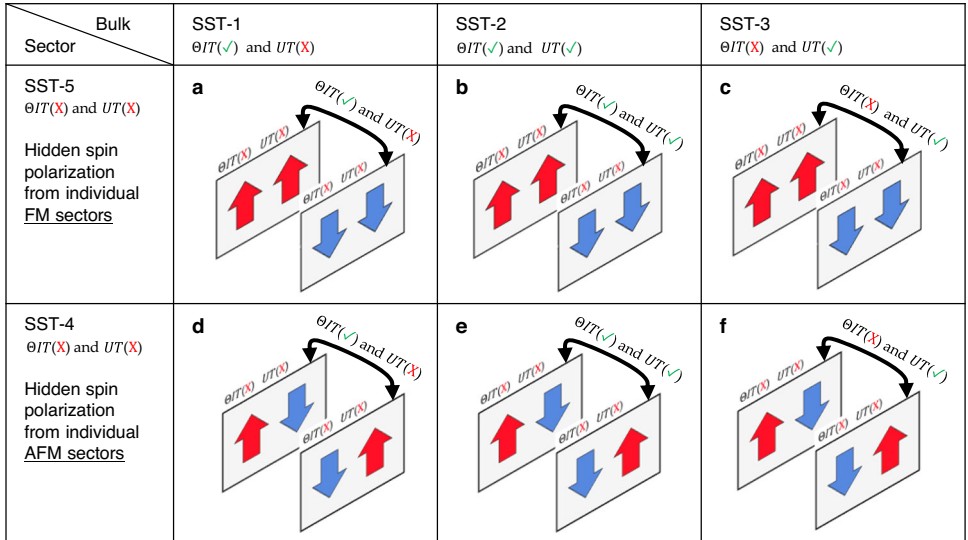

**Fig. 2 | Six types of SOC-independent magnetic hidden spin polarization in collinear antiferromagnets.** These antiferromagnets have global symmetry that disallows spin splitting without SOC, but have lower local sector symmetry that allows spin splitting without SOC. Cases (**a**, **b**, **c**) is where hidden spin polarization arise from local ferromagnetic sectors and cases (**d**, **e**, **f**) is where the hidden spin polarization arise from local antiferromagnetic sectors. Shaded planes are used to indicate the individual sectors that have neither $\Theta IT$ nor $UT$ symmetry and allow spin splitting in the absence of SOC; Parallel and antiparallel arrows of red and blue within the sector plane are used to indicate the ferromagnetic and anti-ferromagnetic ordering of the sector. Sector symmetry is indicated on top of each plane, and bulk symmetry is indicated by the arrow connecting the two sectors.

The three AFM-induced hidden spin polarization cases can then be denoted as (d) bulk SST-1 sector SST-4; (e) bulk SST-2 sector SST-4, and (f) bulk SST-3 sector SST-4. We note that there are multiple ways to decompose the bulk system into sectors, e.g., the bulk SST-I (I = 1,2,3) might also be decomposed into sector SST-I (I = 1,2,3) (or equivalently SST-I (I = 1,2,3) sectors can be used to build the bulk SST-I (I = 1,2,3) materials), where the local spin polarization of each individual sector is still zero, therefore, are not the focus of this work.

## Compounds that have SOC-independent hidden spin polarization

We now turn to discuss how the enabling symmetries are applied to individual sectors to give magnetic hidden spin polarization effects in real antiferromagnetic materials.

As a first step, we will try to find real materials that falls into the six categories we defined. This can be done straightforwardly by applying the symmetry conditions to filter out candidate materials in existing antiferromagnetic databases. We conducted such filtering for MAGN-DATA database[46] and identified a few antiferromagnetic materials of potential candidates for magnetic hidden spin polarization. The identified candidates are: $Ca_2MnO_4$[47], $CoSe_2O_5$[48] and $Fe_2TeO_6$[49], $K_2CoP_2O_7$[50] and $LiFePO_4$[51] whose bulk prototype is SST-1 with sector prototype of SST-4; $Sr_2IrO_4$[52] whose bulk prototype is SST-2 with sector prototype of SST-4; $SrCo_2V_2O_8$[53] whose bulk prototype is SST-3 with sector prototype of SST-4; $CuMnAs$[54] and $Mn_2Au$[55] whole bulk prototype is SST-1 with sector prototype of SST-5; $FeCl_2$ and $CoCl_2$[56] whose bulk prototype is SST-2 with sector prototype of SST-5; $ErAuGe$[57] whose bulk prototype is SST-3 with sector prototype of SST-5. These materials form the platform for the exploration of the magnetic hidden spin polarization effects.

The opposite design philosophy (the bottom-to-top approach) is to construct layered bulk antiferromagnets with the hidden effect based on two-dimensional (2D) compounds that belong to SST-4 and SST-5 prototypes. By searching through the database of predicted naturally exfoliate 3D Van der Waals materials[58], we find a list of 37 ferromagnetic 2D materials and 6 antiferromagnetic 2D monolayers that can be used as such building blocks (see Tables 1 and 2 for the list). Other predicted and synthesized layered 2D materials are either hypothetical or contradictory to enabling symmetry conditions for AFM spin splitting. Van der Waals compounds with spin splitting not only allow the potential practical controllability through external electric fields but also a platform to explore the coexistence of Van der Waals materials properties and AFM-induced spin splitting.

The next step is to validate the predicted hidden spin polarization effect in some of these identified real materials. We studied the sector-projected spin textures on certain wavevector planes for three actual antiferromagnetic materials, $CuMnAs$[54], $Ca_2MnO_4$[47] and $FeBr_2$[56] using PBE + U method[59] in the zero SOC limit. The results are presented below. Additional examples with DFT results are presented in Supplementary Information Section C. These examples proof the existence of the hidden spin polarization effect.

**Hidden spin polarization from individual ferromagnetic sectors.** Figure 3 illustrates the hidden spin polarization effect in tetragonal $CuMnAs$[54] (bulk belonging to SST-1 class with sectors belonging to SST-5 class). The crystal is antiferromagnetically ordered with its magnetic moments collinearly aligned in the (010) direction. The magnetic space group (MSG) of the crystal is Pm′mn (MSG type III). The unit cell consists of two MnAs layers (α-sector and β-sector) that are ferro-magnetically ordered (Fig. 3a, red and blue color shaped polyhedral are used to indicate oppositely magnetized motifs centered on the magnetic sites). By considering the bulk antiferromagnets as a combination of two alternating non-centrosymmetric sectors (α-sector and β-sector), the material has been demonstrated as a useful platform for electrically switching[25,29] the antiferromagnetic magnetization using the hidden spin polarization from the SOC segregated on each sector. Here, we point out a different SOC-independent scenario that might also be contributing to the observed electric switching in this material, i.e., the Zeeman effect within each ferromagnetic MnAs layer creates a local spin split state anchored on the layer. The two MnAs layers are connected by the $\Theta IT$ symmetry which restores the spin degeneracy of the bulk and results in a compensated net spin polarization (Fig. 3b). As shown by the reversed blue and red pattern which are used to map the relative magnitude of the spin up and spin down polarization, the hidden spin polarization is non-zero and is compensated by each other. Examples of hidden spin polarization in spin degenerate bulk

**Table 1 | Easily exfoliable 2D magnetic compounds with AFM configuration belonging to SST-4 class**

| Formula | SG | Structure Prototype | $E_g$ (eV) | $E_b$ (meV/Å²) | 3D SG | SDB of 3D | ID SDB |
|---|---|---|---|---|---|---|---|
| FeSe | P4/nmm | FeSe | 0.0 | 22.6 | Cmme | ICSD | 290411 |
| LaBr | P3m1 | ZrCl | 0.6 | 11.7 | R3m | ICSD | 23354 |
| FeO₂ | Pmmm | FeO₂ | 0.0 | 16.3 | Cmcm | COD | 9015156 |
| PrOI | P4mm | PbClF | 0.0 | 14.9 | P4/nmm | COD | 1530611 |
| FeOCl | Pmmn | PeOCl | 0.0 | 14.2 | Pmmn | COD | 1010645 |
| VOBr | Pmmn | FeOCl | 0.0 | 14.7 | Pmmn | ICSD | 27010 |

The formula, space group, 2D structure prototype, DFT-PBE calculated bandgap ($E_g$), and binding energy ($E_b$) are indicated. The last three columns describe for the experimental parent structure: the 3D space group (3D SG), source database (SDB)[58], and the ID in the source database (ID SDB).

antiferromagnets made of spin split ferromagnetic sectors are also illustrated for CoBr₂[56] (bulk belonging to SST-2 with sector belonging to SST-5) and Ca₃Ru₂O₇[60] (bulk belonging to SST-3 with sector belonging to SST-5) in Supplementary Information Section C.

We note that the corresponding hidden spin polarization projected onto α-sector and β-sector, shown in Fig. 3c, are all aligned in the same direction with the magnetization. Thus, the spin remains a good quantum number. However, the magnitude of the projected spin polarization (mapped by color changing continuously from blue to red) may vary depending on the distribution of the degenerate states on the two sectors. For a pair of degenerate states, the sector projected spin polarization is the summed contribution from both states. For example, the hidden spin polarization of the two spin degenerate states evenly distributed on both sector-α and sector-β($1/\sqrt{2}(|\alpha\uparrow\rangle + |\beta\uparrow\rangle$ and $1/\sqrt{2}(|\alpha\downarrow\rangle - |\beta\downarrow\rangle)$ is $(+0.5) + (-0.5) = 0$ when projected onto sector-α or sector-β; while the hidden spin polarization of the two spin-degenerate states segregated on one of the sector ($|\alpha\uparrow\rangle$ and $|\beta\downarrow\rangle$) is 1 when projected onto sector−α and is −1 when projected onto sector.

**Hidden spin polarization from individual antiferromagnetic sectors.** Fig. 4 illustrates the "hidden spin polarization" effect in antiferromagnetic tetragonal Ca₂MnO₄[47] (bulk belonging to SST-1 class with sector belonging to SST-4 class). The crystal is antiferromagnetically ordered with its magnetic moments collinearly aligned in the (001) direction. The MSG of the crystal is I4₁'/a'cd' (MSG type III). The unit cell consists of two layers of MnO₆ octahedral (α-sector and β-sector) that are antiferromagnetically ordered (Fig. 4a, red and blue color polyhedral are used to indicate oppositely magnetized motifs centered on the magnetic sites). The "magnetic mechanism" [6] within each AFM-ordered sector then creates a local spin split state anchored on the layer. The two MnO₄ layers are connected by the ΘIT symmetry which restores the spin degeneracy of the bulk and results in zero net spin polarization (Fig. 4b). However, the corresponding spin texture projected onto the α-sector and β-sector, shown in Fig. 4c, are persistently aligned in the same direction as its magnetization and are compensated to each other (as indicated by the reversed blue and red pattern which are used to map the relative magnitude of the spin up and spin down polarization). Examples of hidden spin polarization in spin degenerate bulk antiferromagnets made of spin split antiferromagnetic sectors are also illustrated for MnS₂[61] (bulk belonging to SST-2 with sector belonging to SST-4) and La₂NiO₄[62] (bulk belonging to SST-3 with sector belonging to SST-4) in Supplementary Information Section C.

**Revealing and tailoring the hidden spin polarization by external electric field.** To demonstrate the symmetry connection between local sectors and the subsequent transition from hidden effect to apparent effect mediated by the breaking of the symmetry connection, we apply in our calculations a perturbative symmetry-breaking external electric field on an antiferromagnetic compound with hidden spin polarization, hexagonal FeBr₂ (DFT settings for applying the electric field is provided in Methods section). The basic building block of the crystal is the ferromagnetically ordered FeBr₂ layer (sector belonging to SST-5 class). The bilayer slab is built by stacking identical FeBr₂ layer with alternating magnetic ordering. The two layers are connected by both ΘIT and UT symmetry, the bilayer hexagonal FeBr₂ (MSG: P_C-3c1) thus belongs to a bulk SST-2 class, featuring a spin degenerate energy band. However, the spin degenerate band structure of the SST-2 class FeBr₂ (Fig. 5a, b) is lifted upon the application of an external electric field perpendicular to the layers ($E_z$) – a transition from SST-2 to SST-4. The spin splitting arises because of the external electric field $E_z$ creates a non-equivalent potential on the sectors and breaks the ΘIT and UT symmetry of the bulk that connects the two layers. DFT calculations for different values of the applied field, inserted in Fig. 5a, show that such splitting is linearly proportional to the applied external electric field, but in opposite spin polarization ordering for the bottom conduction bands and the top valence bands. The linear field-dependent splitting suggests the split states are segregated on either layer (sector). Indeed, spatial distribution of the spin polarized states, Fig. 5b, c, shows the spin-up (red) state $\Gamma_{CB1}$ is dominantly segregated on the α-sector, while the spin-down (blue) state $\Gamma_{CB2}$ is dominantly segregated on the β-sector. Therefore, the hidden effect of two-fold degenerate energy states subspace (when $E_z = 0$) can be traced back to the individual FeBr₂ layers. Because the applied electric field is small, the main characteristic of the observed spin polarization is inherited from the system without electric field. The layer-segregated states shown in Fig. 5b, c is thus a compelling evidence of the relationship between the global property of spin splitting induced by a global electric field and the local spin polarization. We note the hidden spin polarization effect from local "spin-split" sectors has also been recently exemplified and revealed via an electric field in some antiferromagnets[41,63] where external electric field lifts the spin degeneracy. We also note that the layer Hall effect in the even-layered MnBi₂Te₄−in which electrons from the top and bottom layers spontaneously deflect in opposite directions but globally compensate−has been observed with the help of an applied electric field[30]. These examples not only verify our understanding of the hidden effect being intrinsic to the bulk but also suggest an external electric field as an effective knob for modulating the hidden effect.

## Discussion

**The effect of SOC on the predicted hidden spin polarization**
The SOC-independent hidden spin polarization effect persists in the presence of SOC. This is because the effect being inherited from the unusual antiferromagnetic order rather than SOC[38]. Still, it is important to note the inclusion of SOC would modify the energy bands in both non-magnetic materials and magnetic materials[43,64,65]: (1) it reduces the degeneracy of certain bands which may cause additional spin splitting. (2) it mixes the spin polarized states of up and down (so spin is no longer a good quantum number), which results in momentum-dependent spin polarization that are not unidirectionally aligned; (3) it opens a gap for the crossing energy bands with opposite spin polarization. In compounds consist of low-Z elements the SOC-induced effect can be neglected.

**Table 2 | Easily exfoliable 2D magnetic compounds with FM configuration belonging to SST-5 class**

| Formula | SG | Structure Prototype | $E_g$ (eV) | $E_b$ (meV/Å²) | 3D SG | SDB of 3D | ID SDB |
|---|---|---|---|---|---|---|---|
| $CoBr_2$ | P3m1 | $CdI_2$ | 0.2 | 16.8 | P3m1 | COD | 9016149 |
| $CoCl_2$ | P3m1 | $CdI_2$ | 0.2 | 10.7 | P3m1 | COD | 9014719 |
| $CoO_2$ | P3m1 | $CdI_2$ | 0.0 | 22.6 | P3m1 | COD | 1522027 |
| $FeBr_2$ | P3m1 | $CdI_2$ | 0.0 | 15.5 | P3m1 | COD | 9009102 |
| $FeI_2$ | P3m1 | $CdI_2$ | 0.0 | 16.9 | P3m1 | COD | 9009103 |
| $NiBr_2$ | P3m1 | $CdI_2$ | 0.8 | 18.1 | R3m | COD | 9008013 |
| $NiCl_2$ | P3m1 | $CdI_2$ | 1.1 | 16.3 | R3m | COD | 9009132 |
| $NiI_2$ | P3m1 | $CdI_2$ | 0.3 | 21.5 | R3m | COD | 9009133 |
| $VS_2$ | P3m1 | $CdI_2$ | 0.0 | 27.7 | P3m1 | ICSD | 651361 |
| $VSe_2$ | P3m1 | $CdI_2$ | 0.0 | 25.4 | P3m1 | ICSD | 86520 |
| $VTe_2$ | P3m1 | $CdI_2$ | 0.0 | 27.1 | P3m1 | ICSD | 603582 |
| $TmI_2$ | P3m1 | $CdI_2$ | 0.0 | 10.5 | P3m1 | ICSD | 43731 |
| $LaBr_2$ | P6m2 | $MoS_2$ | 0.6 | 11.2 | P6₃/mmc | ICSD | 65481 |
| FeTe | P4/nmm | FeSe | 0.0 | 26.6 | P4/nmm | ICSD | 169974 |
| LaCl | P3m1 | ZrCl | 0.0 | 11.0 | R3m | COD | 24410 |
| ScCl | P3m1 | ZrCl | 0.0 | 13.8 | R3m | COD | 4343683 |
| TbBr | P3m1 | ZrCl | 0.0 | 12.2 | R3m | ICSD | 23353 |
| YCl | P3m1 | ZrCl | 0.0 | 17.6 | R3m | ICSD | 30708 |
| $CuCl_2$ | C2/m | $NbTe_2$ | 0.2 | 13.4 | C2/m | COD | 9001506 |
| EuOBr | P4/nmm | PbClF | 0.0 | 17.4 | P4/nmm | ICSD | 28531 |
| EuOI | P4/nmm | PbClF | 0.0 | 14.8 | P4/nmm | ICSD | 27666 |
| PrOBr | P4/nmm | PbClF | 0.0 | 24.1 | P4/nmm | COD | 2232654 |
| NdOBr | C2/m | PbClF | 0.2 | 21.8 | P4/nmm | COD | 9009172 |
| SmOBr | C2/m | PbClF | 0.2 | 18.4 | P4/nmm | COD | 1530050 |
| TmOI | C2/m | PbClF | 0.2 | 15.0 | P4/nmm | COD | 2310429 |
| TbOBr | Cmme | PbClF | 0.0 | 15.2 | P4/nmm | ICSD | 28532 |
| CrOBr | Pmmn | FeOCl | 0.5 | 14.8 | Pmmn | ICSD | 27092 |
| CrOCl | Pmmn | FeOCl | 0.6 | 13.8 | Pmmn | ICSD | 4086 |
| CrSBr | Pmmn | FeOCl | 0.4 | 19.5 | Pmmn | ICSD | 69659 |
| ErSCl | Pmmn | FeOCl | 0.3 | 11.9 | Pmmn | ICSD | 21009 |
| ErSeI | Pmmn | FeOCl | 0.0 | 11.6 | Pmmn | ICSD | 50194 |
| HoSI | Pmmn | FeOCl | 0.5 | 10.9 | Pmmn | ICSD | 425295 |
| ErHCl | P3m1 | SmSI | 0.0 | 10.9 | R3m | COD | 1530725 |
| SmSI | P3m1 | SmSI | 0.0 | 11.4 | R3m | COD | 1008317 |
| YbOCl | P3m1 | SmSI | 0.0 | 11.7 | R3m | ICSD | 6077 |
| CdOCl | P3m1 | BiTeI | 0.3 | 25.6 | P6₃mc | COD | 9016472 |
| $Co(OH)_2$ | C2/m | $Mg(OH)_2$ | 0.0 | 18.3 | P3m1 | ICSD | 88940 |

The formula, space group, 2D structure prototype, DFT-PBE calculated bandgap ($E_g$), and binding energy ($E_b$) are indicated. The last three columns describe for the experimental parent structure: the 3D space group (3D SG), source database (SDB)[58], and the ID in the source database (ID SDB).

## Use magnetic symmetry with SOC to describe the spin-splitting of energy bands without SOC

In collinear antiferromagnetic compounds, the existence of $UT$ in the spin space group (SSG, symmetry group of the system without SOC) means there is a spatial translation $T$ that connects the atomic sites with opposite magnetic moments and keeps the crystal structure invariant. By definition, antiferromagnets with primitive lattice translations that reverse the microscopic magnetic moments are known as having black and white Bravais lattice that is classified as MSG type IV; Antiferromagnets without such translation $T$ belongs to MSG type I and type III[66]. This suggests there is a one-to-one correspondence between the existence or absence of the $UT$ in the spin space group and the MSG being type IV or type I/III.

The correspondence relation can be formally established by introducing an auxiliary MSG—a subgroup of the spin space group containing only elements of spatial and time reversal symmetries. This is referred to as "MSG without SOC" in the Appendix of ref. [38] or equivalently as "magnetic groups with pseudoscalar electron spin" in ref. [67]. Following that, we can prove a chain relation as depicted in Eq. (1).

$$\begin{aligned}&\text{(a)}\ UT\ \text{in SSG} \leftrightarrow \text{(b)}\ \Theta T\ \text{in Auxiliary MSG} \leftrightarrow \\ &\text{(c)}\ \Theta T\ \text{in Standard MSG} \leftrightarrow \text{(d)}\ \text{Standard MSG being type IV}\end{aligned} \quad (1)$$

(a) The existence or not of a $UT$ symmetry in the SSG corresponds to (b) the existence or not of a $\Theta T$ symmetry in the auxiliary group. This is because the $\Theta U$ symmetry preserves any collinear magnetic ordering and is a symmetry of any collinear antiferromagnets[43–45]. Meanwhile, (b) the existence or not of a $\Theta T$ symmetry in the auxiliary MSG (without SOC) corresponds to (c) the existence or not of a $\Theta T$ symmetry in the standard MSG (with SOC). Antiferromagnetic materials whose (c) MSG preserve (or violate) $\Theta T$ symmetry is classified as (d) MSG type IV (or MSG type I/III)[66].

The established correspondence relation thus justifies the use of MSG (with SOC)—avoiding the use of the "less familiar" spin symmetry[64]—to predict whether the spin splitting effect without SOC will occur. This also allows the use of the tabulated magnetic structure symmetry information provided in material database[46] to sort out candidate materials[39]. For the prediction of the degeneracy of the full bands without SOC, a comprehensive analyze of the spin symmetry group and its irreducible representation is necessary[43,64,65].

## Hidden versus apparent spin polarization in noncollinear antiferromagnets

While the current paper focuses on the hidden spin polarization in collinear antiferromagnetic compounds, we note that the hidden effect can also exist in noncollinear antiferromagnetic compounds. When a bulk noncollinear antiferromagnetic compound has $\Theta IT$ symmetry, the energy bands are spin degenerate. If the system can be further divided into separate sectors that locally violate $\Theta IT$, then there could exist hidden spin polarization pertaining to the individual sectors. However, one should note that the symmetry condition of having $UT$ for preserving spin degeneracy in noncollinear antiferromagnetic compounds[39] is not valid anymore, this is because (1) when the spin arrangement is non-coplanar, the MSG type IV does not guarantee the existence of $UT$; Moreover, (2) when the spin arrangement is coplanar, MSG type IV guarantees the existence of $UT$, but the existence of such $UT$ does not always guarantee spin degeneracy. Specifically, when the spin states are not aligned in the same plane of the coplanar plane, the $UT$ symmetry will not reverse the spin states as it works in the collinear magnetic systems. These properties of noncollinear antiferromagnets offer new knobs to tune the hidden versus apparent spin polarization via tilting the local magnetic motifs.

## Experimental detectability

Analogous to the detection of SOC-induced hidden spin polarization in nonmagnetic compounds (also known as R-2 and D-2 effects) [12], a hidden property can be observed when a probe can resolve the local sectors where the property is not compensated. Specific to hidden spin polarization, the spatial segregation of the spin polarization states allows in principle the detection of the hidden effect in

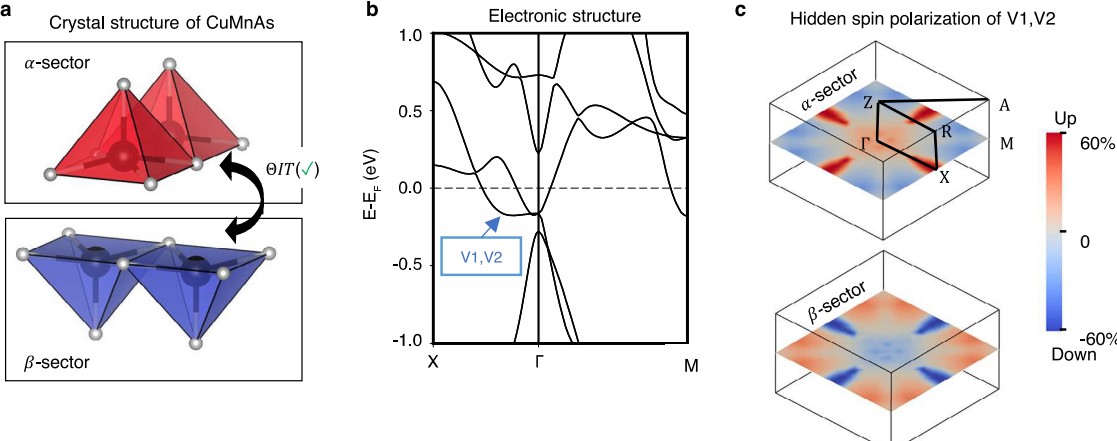

**Fig. 3 | Hidden spin polarization from individual ferromagnetic sectors in bulk tetragonal CuMnAs (bulk belonging to SST-1 class with sector belonging to SST-5 class). a** Crystal structure of antiferromagnetic CuMnAs composed of two ferromagnetic layers with opposite magnetization (indicated by red and blue polyhedra) in the unit cell. The Cu atoms are dismissed. The two layers are referred to as sector-$\alpha$ and sector-$\beta$, respectively; (**b**) Spin degenerate band structure of CuMnAs; (**c**) Hidden spin polarization from each individual sector of the highest two valence bands (V1 and V2) on $\Gamma$XS k-plane. The up and down spins are mapped to the color from blue to red. The crystal and magnetic structure for tetragonal CuMnAs used in our DFT calculations are taken from ref. [54].

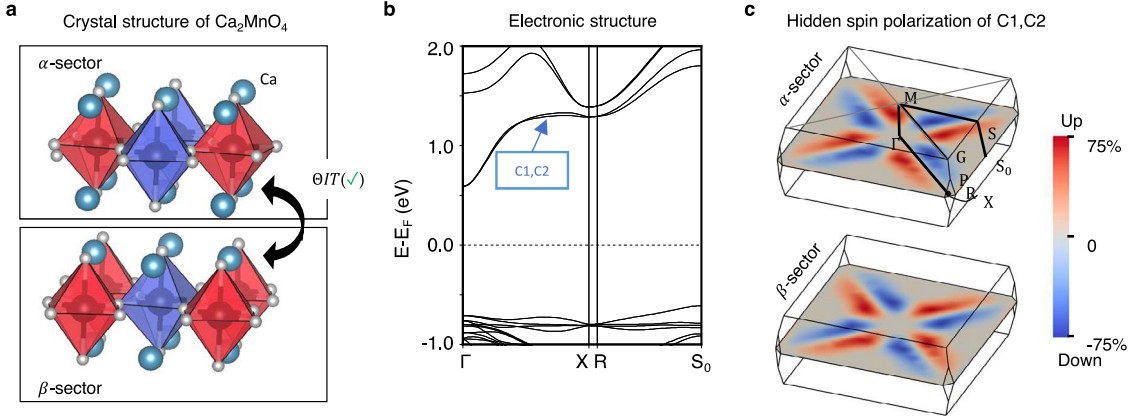

**Fig. 4 | Hidden spin polarization from the individual antiferromagnetic sector in bulk tetragonal Ca$_2$MnO$_4$ (bulk belonging to SST-1 class with sector belonging to SST-4 class). a** Crystal structure of antiferromagnetic tetragonal Ca$_2$MnO$_4$ composed of two antiferromagnetic sectors with opposite magnetic ordering (the magnetic ordering is indicated by red and blue polyhedra) in the unit cell. The two layers are referred to as sector-$\alpha$ and sector-$\beta$, respectively; (**b**) Spin degenerate band structure of Ca$_2$MnO$_4$; (**c**) Hidden spin polarization from each individual sector of the lowest two conduction bands (C1 and C2) on $\Gamma$XR k-plane. The up and down spins are mapped to the color from blue to red. The crystal and magnetic structure for tetragonal Ca$_2$MnO$_4$ used in our DFT calculations are taken from ref. [47].

antiferromagnets. Since this effect is intrinsic to the bulk it can be distinguished from the surface effect as the latter sensitively depends on the effective penetration depth of the probing beam [43]. Albeit, for the hidden spin polarization from individual AFM sectors, to detect the AFM spin polarization of the individual sectors, one needs to choose the surface configuration that respects the symmetries of the individual sector that ensure the anti-ferromagnetism of the sector, e.g., mirror plane symmetries perpendicular to the surface plane that connect the spin up and spin down magnetic moments of the AFM sector. Especially, systems with the degenerate states segregated on the different sectors would result in a minimally compensated hidden spin polarization, thus contributing to a robust signal when selectively probing the individual sector, thus being ideal platforms for the detection of the hidden effect.

### Electric and magnetic field control of the hidden effect
One of the most desirable features of spin-related phenomena is the possibility of electric and magnetic control. In the case of the hidden spin polarization in AFM, since the unit cell can always be built in terms of two or more sectors, electric field is a practically direct way of inducing and controlling the existence of spin splitting (as well as its magnitude) via modulating the symmetry relationship between the sectors. For example, in the spin degenerate bulk antiferromagnets made of a pair of spin-split antiferromagnetic sectors (e.g., FeSe discussed in Supplementary Information Section D) or ferromagnetic sectors (e.g., FeBr$_2$ discussed in the Results Section), external electric field would break the $\Theta IT$ and $UT$ symmetry between the spin split sectors, which then implies a transition from hidden effect to apparent effect. In fact, the electric field applied couples with the electron spin through the magnetoelectric effect[68], which is only allowed under specific symmetry conditions[41]. Additionally, transport properties that are even functions of the sectors can take non-vanishing values in a hidden system. For example, non-reciprocal nonlinear current respond to an applied electric field is recently demonstrated in antiferromagnetic tetragonal CuMnAs[69]. This serves as a guide in search for systems exhibiting this particular

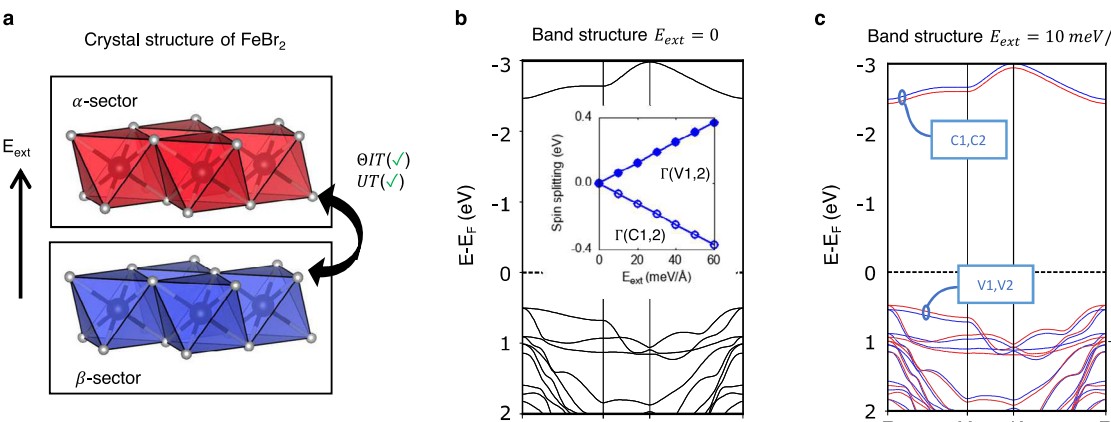

**Fig. 5 | Revealing the hidden spin polarization in hexagonal FeBr₂ using an external electric field. a** spin split band structure of FeBr₂ with a 10 meV/Å z-oriented external electric field. Red and blue lines represent the spin-up and spin-down polarized bands. Insert depicts the spin splitting between the bottom two conduction bands at Γ (denoted as $\Gamma_{CB1}$ and $\Gamma_{CB2}$) as a function of the external electric field; (**b**) wavefunction plot for $\Gamma_{CB1}$; and (**c**) wavefunction plot for $\Gamma_{CB2}$. The crystal and magnetic structure for triagonal FeBr₂ are taken from ref. [56]. and was tailored into a bilayer slab for the calculations with external electric field.

response behavior. Furthermore, the bulk antiferromagnets formed by ferromagnetic layers with alternatively aligned magnetic moments along the direction perpendicular to the ferromagnetic layers (thus hosting hidden spin polarization) could have very different magnetoresistance from the bulk ferromagnets formed by the same ferromagnetic layers but with uniformly aligned magnetic moments. Therefore, switching between the AFM and FM states by external magnetic field could lead to significant change of magnetoresistance, mimicking the tunneling magnetoresistance effect[70]. These perspectives offer electric and/or magnetic means to control the spin-related properties in antiferromagnets.

## Methods
### DFT setup
Electronic structures are calculated using the density functional theory (DFT) method[71–73] with the General Gradient Approximation (GGA)[74,75] implemented in the Vienna Ab initio simulation package (VASP). Structural and magnetic configurations are taken from the MAGN-DATA database[46] derived from experiments. The calculations of SOC-independent spin splitting and spin polarization are done using a non-collinear magnetic setting but without the introduction of spin-orbit coupling (i.e., SOC turned off). We adopt the GGA + U method[76] to account for the on-site Coulomb interactions of localized 3d orbitals involved in the calculations. We used $U = 3.9$ eV, $J = 0$ eV on Mn-3d orbits for Ca₂MnO₄, $U = 5.3$ eV, $J = 0$ eV on Fe-3d orbits for insulating FeBr₂. These Hubbard $U$ values are derived in ref. [77] using the approach outlined in ref. [78]. For CuMnAs, a metal, we used a smaller $U$ value on Mn-3d orbits ($U = 1.9$ eV, $J = 0$ eV[39]). We follow the approach proposed by Neugebauer and Scheffler[79] to apply a uniform electric field to the bilayer slab in the calculations. This approach treats the artificial periodicity of the slab by adding a planar dipole sheet in the middle of the vacuum region.

### How is the hidden spin polarization calculated
We evaluated the hidden spin polarization on sector-η by projecting the calculated degenerate wavefunctions $|\phi_1\rangle$, $|\phi_2\rangle$ onto the atomic orbital basis $|ilm\rangle$ and sum over the sites i within sector-η in the primitive unit cell, $S_\eta = \sum_{s=1,2}\sum_{i\in\eta}\sum_{lm}\langle\phi_s|\hat{S}|ilm\rangle\langle ilm|\phi_s\rangle$. This expression sums contribution from both degenerate bands ($s = 1,2$).

### How are "sectors" chosen
Sectors are chosen such that atomic sites within a sector are more closely clustered, while atom pairs associated with different sectors are spatially well separated. This results in weak inter-sector coupling, and consequently physically significant hidden spin polarization effect.

## Data availability

The VASP configuration and output files that support the finding of this study have been deposited in figshare with the identifier [data DOI:10.6084/m9.figshare.22693042]. Other data related to this research are available from the corresponding author upon reasonable request.

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

## Acknowledgements

Theoretical work on formal symmetry of hidden effects at Colorado University Boulder was supported by the National Science Foundation (NSF) DMR-CMMT Grant No. DMR-2113922. The electronic structure calculations of this work were supported by the US Department of Energy, Office of Science, Basic Energy Sciences, Materials Sciences and Engineering Division under Grant No. DE-SC0010467. This work used resources of the National Energy Research Scientific Computing Center, which is supported by the Office of Science of the US Department of Energy. This work also used resources of Stampede2 system at Texas Advanced Computing Center through allocation PHY180030 from the Advanced Cyberinfrastructure Coordination Ecosystem: Services & Support (ACCESS) program, which is supported by National Science Foundation grants #2138259, #2138286, #2138307, #2137603, and #2138296.

## Author contributions

A.Z. conceived and supervised the project. L.-D.Y., X.Z. and C.M.A. developed the idea and designed the research. L.-D.Y. and X.Z. conducted the search for candidate antiferromagnetic materials from the existing database. L.-D.Y. performed the density functional calculations and plot the Figures. C.M.A. performed the "bottom-to-top" material design of layered bulk antiferromagnets with the hidden effect. L.-D.Y., X.Z., C.M.A., and A.Z. analyzed the results and wrote the manuscript.

## Competing interests

The authors declare no competing interests.
