## [Peer Review File · Nature Communications]

Reviewers' Comments:

Reviewer #1:

Remarks to the Author:

Yuan et al. report calculations of local spin polarizations in antiferromagnets which is unrelated to spin-orbit coupling. The local spin polarisations are calculated using density functional theory and are referred as "hidden" effect by the authors. The work is an extension of the Phys. Rev. Materials 5, 014409 by some of the present authors (Ref 30). In the current manuscript, they focus on the PROJECTED spin polarizations in antiferromagnets with GLOBALLY spin-degenerate bands. They study two types of local spin polarization, "ferromagnetic" (CuMnAs, also labeled prototype SST-5) and antiferromagnetic (Ca₂MnO₄, SST-4). The authors argue that some of the local spin polarizations can induce effects that can be measured by applying an electric field to the system (FeBr₂), and may be relevant to previously observed effects in antiferromagnets, such as spin-polarized surface states in NdBi, or electric switching of CuMnAs. The authors chose a very active area of antiferromagnetic spintronics and also some relevant materials studied within this area, such as CuMnAs. The result may be of potential interest to the subcommunity of condensed matter physics concerned with antiferromagnets. The possibility of ferromagnetic or antiferromagnetic local spin polarization in spin-degenerate antiferromagnets - appears to be in principle valid. However, the full and quantitative validity of the results cannot be assessed because some important methodology information is missing from the manuscript, such as the formulas used to calculate the spin projections and calculation parameters (electron correlations strength), which make it impossible to reproduce the results. Also some relevant literature on "hidden effects" (such as Nature 595, 521 (2021), Phys. Rev. B 102, 125123 (2020)) does not seem to be properly cited and thus the level of novelty of the current manuscript is not clear. In addition, the argued relevance of the hidden effects for previously reported experiments in NdBi and for switching antiferromagnets needs to be either explained in more detail or removed. The readability of the manuscript for the broad readership of Nature Communications is also hampered by the introduction of many, in my opinion, unnecessary and possibly misleading terms (such as "centrosymmetric antiferromagnets", "SST-prototypes"). I explain all points in detail below.

In summary, I cannot recommend the paper in its current form for publication in Nature Communications. However, I can foresee eventual reconsideration if the enclosed comments and questions are addressed.

Detailed comments and questions

1. The authors should explain the calculation details of sector spin-polarisations.
 - a. What formula is used to project the spin polarization onto a given crystallographic sector and how is the sector chosen (which atoms)?
 - b. What is the numerical scale in the colour plots of spin-polarisations of Figures 3 and 4? Does it correspond to the summed contribution from both of the Kramer's degenerate bands? When the spin-orbit coupling is switched off in the calculations, the spin should be a good quantum number. Why is it not the case in plots such as in Figure 3 (c)?
2. Could the authors provide material parameters (e.g., Hubbard U)? Could they also plot the band structure along the complete conventional momentum space path and comment on the comparison of the obtained band structure with those reported in the literature? It is not clear what bands are plotted in Figure 3 (c), because in Figure 3 (b) several different bands cross the Brillouin zone.
3. The effect of spin-orbit interaction on the calculated energy bands and spin polarizations should be discussed.
 - a. The spin-orbit interaction cannot be turned off in real materials. How do the calculated projected spin polarizations given in the main text change when the spin-orbit interaction is included?
 - b. The authors argue that it is legitimate to use magnetic space groups with SOC to describe the spin polarisation of energy bands WITHOUT SOC. However, it is well-known that even energy bands differ when calculated with and without SOC. This is well established for paramagnetic materials and for magnetic materials it was discussed e.g. in Ref. 50 and Phys. Rev. X 12, 021016 (2022). Could the authors adapt this statement?

4. The authors claim that effects observed despite enabling symmetry being absent can be explained by the "hidden effects" (e.g. lines 37-39, and 44-45). However, the authors do not prove the relationship between local projected spin polarization and observable global effects, e.g. due to an applied electric field. In fact, they show that induced spin splitting in FeBr₂ is proportional to electric field gating strength, which is not projection/sector dependent, but rather a global property. Could the authors find more compelling evidence of the relationship between the global property of spin splitting induced by a global electric field and the local spin polarization?

5. The authors should clarify their claims regarding the explanation of the observed spin polarization in NdBi and the manipulation of antiferromagnetism in CuMnAs.

a. In NdBi, Ref.51, spin polarization is reported in the surface spectrum/Fermi arcs, while the present authors focus on the bulk spectrum. How are the results of the bulk spectrum relevant to the surface spectrum?

b. The quantization axes for local spin polarization and magnetic ordering are the same, and therefore it is not clear how the local spin polarization unrelated to spin-orbit coupling can torque the magnetic ordering itself. The claims in lines 200-202 about CuMnAs are thus not justified.

6. Some relevant references to previous work on local/global spin effects in antiferromagnets are omitted:

a. How does the concept of "hidden sector" effects in this manuscript differ from the concepts of "hidden layer" effects discussed in Nature 595, 521 (2021) and references therein? It appears that the electric field-induced spin polarization in FeBr₂ (the same prototype as CuMnAs) is conceptually similar to the electric field-induced spin polarization in MnBi₂Te₄, see Figure 3g in Nature 595, 521 (2021).

b. How do the present results compare to the local sector spin-polarisations calculated in CuMnAs previously, Phys. Rev. B 102, 125123 (2020).

c. When citing "centrosymmetric antiferromagnets" (e.g. line 91), also some previous works on Kramers degeneracy in CuMnAs with combined space inversion and time-reversal symmetry should be cited: Phys. Rev. Lett. 118, 106402 (2017) and Phys. Rev. B 102, 125123 (2020)

d. Spin splitting in antiferromagnets has also been reported in Phys. Rev. B 99, 184432 and Sci. Adv. 6 : eaaz8809 (unrelated to SOC), and in Phys. Rev. B 100, 245115 (related to SOC). The authors even use in Table in panel b of Figure 1 the BiCoO₃ and rutile crystals from these references as material examples.

7. The paper introduces some terminology, which, however, might be confusing for the general readership of Nature Communications:

a. The topic of the paper is the local spin polarisation in antiferromagnets and in principle only two prototypes are discussed - ferromagnetic (Figure 3/5) and antiferromagnetic (Figure 4). However, in Figure 1 the authors repeat the results/terminology from Phys. Rev. Materials 5, 014409 for global spin polarisation without citing it in the caption. Moreover, although the panel in Figure 1c is labelled as spin polarization sectors and refers to local spin polarisations of spin-degenerate antiferromagnets, they give examples of globally spin-polarized antiferromagnets/ferromagnets.

b. The authors refer to CuMnAs and Mn₂Au as "centrosymmetric antiferromagnets", which can be misleading. If I understand correctly, the authors mean by this terminology that a CuMnAs/Mn₂Au crystal in the paramagnetic state has an inversion, but not in the antiferromagnetic state (see also point 6 c.).

8. Some of the typos should be corrected, such as "Dresslhouse" (line 50), or missing characters in the caption of supplementary Figure S5 (lines 114-115).

Reviewer #2:

Remarks to the Author:

In the manuscript under consideration, the authors use a "top-down" strategy to engineer antiferromagnetic materials featuring a hidden spin-polarization. The idea works in the following way: a careful analysis of the underlying symmetries suggests that sectors characterized by apparent (i.e. visible) spin-polarization can be stacked together, in a compensated way, in order to feature a hidden spin-polarization in the bulk. By hidden spin-polarization, the authors meant a

finite spin expectation value when the Bloch wave functions are projected onto the individual sectors (layers). What I find interesting in this work is that the spin-polarization is achieved without the requirement of spin-orbit coupling (SOC). Indeed, the spin-polarization of each sector is achieved either by AFM or FM ordering. Driven by the symmetry analysis, the authors then scrutinized a vast database of known collinear AFM materials, and performed first-principles calculations on several candidate compounds to bring evidence of the solidity of their “top-down” strategy.

This work represents an important step further beyond the already known hidden Rashba and Dresselhaus spin-splittings that unavoidably require a sizable contribution from SOC. Here, on the other hand, the concept of “hidden” is extended to low-Z compounds, for which SOC is most of the times a tiny perturbation. I find the proposal convincing and potentially relevant to further boost the theoretical and experimental efforts in searching for new spin-polarized materials. I thus recommend its publication in Nature Communications.

Reviewer #3:

Remarks to the Author:

The manuscript by Lin-Ding Yuan, Xiuwen Zhang, Carlos Mera, and Alex Zunger is devoted to an interesting feature of electron energy bands in collinear antiferromagnets with negligible spin-orbit interaction, namely to the spin polarization of the bands which can be revealed only after a projection on different sectors of the systems is carried out. The authors call this phenomenon a 'hidden' spin polarization and put it in a broader context of effects that occur in a system due to its symmetry. Based on the symmetry arguments, the authors define six different types of the hidden spin polarization, identify potential candidate materials exhibiting the effect by filtering a database of magnetic materials, and discuss the response of the electron bands to external electric fields as well as the potential applicability. The authors' statements are illustrated by figures presenting results of electronic-structure calculations for selected compounds, obtained by up-to-date techniques.

On one hand, the essence of the phenomenon is nearly trivial, as can easily be found by inspecting the electron bands of the simplest one-dimensional model of an antiferromagnet (with two sites in the unit cell, described with a single orbital per site and a spin-independent nearest-neighbor hopping). On the other hand, however, the systematic classification of various antiferromagnets and their sectors (ferro- or antiferromagnetic), and the outlined relevance for the whole area of spintronics will undoubtedly attract attention both by theoreticians and by experimentalists working in the field. It should also be noted that collinear antiferromagnets with weak spin-orbit interaction are intensively studied at present, as can be documented, e.g., by a very recent review Smejkal et al., Phys. Rev. X 12, 040501 (2022). For these reasons, I believe that the authors' manuscript deserves publication in a reputable scientific journal.

In my view, however, the work can be improved in two directions:

1) Proper characterization of the underlying symmetry groups.

The authors use the standard magnetic space groups (Shubnikov space groups, relevant for electron systems with spin-orbit interaction) for a treatment of systems without spin-orbit interaction. This seems advantageous over approaches based on 'less familiar' and more exotic spin groups featured by two independent rotations (in the configuration space and in the spin space). However, their formalism employs an operation U (a spin rotation acting only in the spin space that causes a spin reversal) which does not belong among the basic operations of the traditional magnetic groups. It seems to me that this addition of U to the other symmetry operations moves the actual group formalism towards the spin groups. Readers would like to know whether the complete group (i.e., the magnetic space group extended by the spin-only operation U) is equivalent to the full spin group, or to its subgroup, or whether the authors introduce implicitly a new kind of groups.

On a more pragmatic level, one would like to know how the filtering using the MAGNDATA

information (with the symmetry of each system defined by the magnetic space group) was carried out. I think that the presence/absence of the spatial translations (T), time reversal (theta), the spatial inversion (I), and their combinations can be obtained in a straightforward manner (in principle, just by looking at the list of elements of the magnetic space group). However, the way to assess the presence/absence of U (or UT) among the symmetry elements of the system only from its magnetic space group alone is unclear to me.

2) Relation of the introduced classification to the response behavior.

The various kinds of the response of a bulk system to an external electric field lead to well-known quantities, such as the charge and spin conductivity, but also to novel quantities, such as the current-induced spin torques or current-induced spin polarization. I wonder whether the authors can establish a relation between the six different types of the collinear antiferromagnets and the vanishing/nonvanishing values of a transport response quantity. This could serve as a guide in search for systems exhibiting that particular response behavior and it would underline the importance of the introduced classification scheme.

I have also found two minor points to be corrected:

3) In the caption to Figs. 3 and 4 (and also to Figs. S1 - S4 of the Supplement), the expression 'polynomials' should perhaps be replaced by 'polyhedra'.

4) In the text of the Supplement, in the parts describing the results for La_2NiO_4 and MnS_2 systems, one can read 'The two ferromagnetically ordered Ru_2O_7 sectors are ...', which should be corrected.

In summary, I find the submitted work interesting and potentially important for the field of spintronics. Nevertheless, the work can be improved along the indicated directions, which could strengthen its impact on the physical community.

A point-by-point response to the referees' comments:

Reviewer #1 comments and author response

Referee Comment 1: “Yuan et al. report calculations of local spin polarizations in antiferromagnets which is unrelated to spin-orbit coupling. The local spin polarisations are calculated using density functional theory and are referred as "hidden" effect by the authors. The work is an extension of the Phys. Rev. Materials 5, 014409 by some of the present authors (Ref 30). In the current manuscript, they focus on the projected spin polarizations in antiferromagnets with globally spin-degenerate bands. They study two types of local spin polarization, “ferromagnetic” (CuMnAs, also labeled prototype SST-5) and antiferromagnetic (Ca₂MnO₄, SST-4). The authors argue that some of the local spin polarizations can induce effects that can be measured by applying an electric field to the system (FeBr₂), and may be relevant to previously observed effects in antiferromagnets, such as spin-polarized surface states in NdBi, or electric switching of CuMnAs. The authors chose a very active area of antiferromagnetic spintronics and also some relevant materials studied within this area, such as CuMnAs. The result may be of potential interest to the subcommunity of condensed matter physics concerned with antiferromagnets. The possibility of ferromagnetic or antiferromagnetic local spin polarization in spin-degenerate antiferromagnets - appears to be in principle valid. However, the full and quantitative validity of the results cannot be assessed because some important methodology information is missing from the manuscript, such as the formulas used to calculate the spin projections and calculation parameters (electron correlations strength), which make it impossible to reproduce the results. Also some relevant literature on "hidden effects" (such as Nature 595, 521 (2021), Phys. Rev. B 102, 125123 (2020)) does not seem to be properly cited and thus the level of novelty of the current manuscript is not clear. In addition, the argued relevance of the hidden effects for previously reported experiments in NdBi and for switching antiferromagnets needs to be either explained in more detail or removed. The readability of the manuscript for the broad readership of Nature Communications is also hampered by the introduction of many, in my opinion, unnecessary and possibly misleading terms (such as "centrosymmetric antiferromagnets", "SST-prototypes"). I explain all points in detail below. In summary, I cannot recommend the paper in its current form for publication in Nature Communications. However, I can foresee eventual reconsideration if the enclosed comments and questions are addressed.”

Author Response: We have been able to incorporate changes in the manuscript to address these concerns. Specifically, we have added in **Methods section** the quantitative formula we used in calculating the “spin projections” and material-specific “calculation parameters” used to generate our results. We have commented/discussed the relevant literature, pointed out by the reviewer, on their connections and differences to the current manuscript in **Introduction section**. We note the relevant previous works pointed out by referee #1 consider SOC to be a prerequisite in one or two specific materials are relevant but conceptually different from the current work where we performed systematic classification and exploration of the SOC independent hidden effects. We have also provided definitions or clarifications for the ambiguous terms where they were first introduced. Reply and changes made to the manuscript to address the specific comments and questions are given below.

Referee Comment 2: Detailed comments and questions

1. The authors should explain the calculation details of sector spin-polarisations.

a. What formula is used to project the spin polarization onto a given crystallographic sector and how is the sector chosen (which atoms)?

b. What is the numerical scale in the colour plots of spin-polarisations of Figures 3 and 4?

Does it correspond to the summed contribution from both of the Kramer's degenerate bands?

When the spin-orbit coupling is switched off in the calculations, the spin should be a good quantum number. Why is it not the case in plots such as in Figure 3 (c)?

Author Response:

To address (a), we have added a description of how we calculate the ‘projected’ hidden spin polarization, and how we choose the sector in **Methods section (page 14)**:

“How is the hidden spin polarization calculated: We evaluated the hidden spin polarization on sector- η by projecting the calculated degenerate wavefunctions $|\phi_1\rangle, |\phi_2\rangle$ onto the atomic orbital basis $|ilm\rangle$ and sum over the sites i within sector- η in the primitive unit cell, $S_\eta = \sum_{s=1,2} \sum_{i \in \eta} \sum_{lm} \langle \phi_s | \hat{S} | ilm \rangle \langle ilm | \phi_s \rangle$. This expression sums the contribution from both degenerate bands ($s=1,2$).

How are “sectors” chosen: Sectors are chosen such that atomic sites within a sector are more closely clustered, while atom pairs associated with different sectors are spatially well separated. This results in weak inter-sector coupling, and consequently physically significant hidden spin polarization effect.

To address (b), we have edited Figures 3 and 4 to specify the numerical scale of the color plots of spin polarizations. We have also provided additional clarifications to the “binary color range” representation of the projected “hidden spin polarization” (**page 8**):

“We note that the corresponding hidden spin polarization projected onto α -sector and β -sector, shown in Fig. 3c, are all aligned in the same direction with the magnetization. Thus, the spin remains a good quantum number. However, the magnitude of the projected spin polarization (mapped by color changing continuously from blue to red) may vary depending on the distribution of the degenerate states on the two sectors. For a pair of degenerate states, the sector projected spin polarization is the summed contribution from both states. For example, the hidden spin polarization of the two spin degenerate states evenly distributed on both sector- α and sector- β ($1/\sqrt{2}(|\alpha \uparrow\rangle + |\beta \uparrow\rangle)$ and $1/\sqrt{2}(|\alpha \downarrow\rangle - |\beta \downarrow\rangle)$) is $(+0.5)+(-0.5)=0$ when projected onto sector- α or sector- β ; while the hidden spin polarization of the two spin-degenerate states segregated on one of the sector ($|\alpha \uparrow\rangle$ and $|\beta \downarrow\rangle$) is 1 when projected onto sector- α and is -1 when projected onto sector- β .”

Figure 3: Hidden spin polarization from individual ferromagnetic sectors in bulk tetragonal CuMnAs (bulk belonging to SST-1 class with sector belonging to SST-5 class). **a** Crystal structure of antiferromagnetic CuMnAs composed of two ferromagnetic layers with opposite magnetization (indicated by red and blue polyhedra) in the unit cell. The Cu atoms are dismissed. The two layers are referred to as sector- α and sector- β , respectively; **b** Spin degenerate band structure of CuMnAs; **c** Hidden spin polarization from each individual sector of the highest two valence bands (V1 and V2) on Γ XS k-plane. The up and down spins are mapped to the color from blue to red. The crystal and magnetic structure for tetragonal CuMnAs used in our DFT calculations are taken from Ref. [55].

Figure 4: Hidden spin polarization from the individual antiferromagnetic sector in bulk tetragonal Ca₂MnO₄ (bulk belonging to SST-1 class with sector belonging to SST-4 class). **a** Crystal structure of antiferromagnetic tetragonal Ca₂MnO₄ composed of two antiferromagnetic sectors with opposite magnetic ordering (the magnetic ordering is indicated by red and blue polyhedra) in the unit cell. The two layers are referred to as sector- α and sector- β , respectively; **b** Spin degenerate band structure of Ca₂MnO₄; **c** Hidden spin polarization from each individual sector of the lowest two conduction bands (C1 and C2) on Γ XR k-plane. The up and down spins are mapped to the color from blue to red. The crystal and magnetic structure for tetragonal Ca₂MnO₄ used in our DFT calculations are taken from Ref. [48].

Referee Comment 3: Could the authors provide material parameters (e.g., Hubbard U)?

Author Response: We have added the material parameters used in our calculations in **Methods section** and corresponding **Figure captions**.

In the **Methods section**, we have added:

“...We used $U=3.9$ eV, $J=0$ eV on Mn-3d orbits for Ca_2MnO_4 , $U=5.3$ eV, $J=0$ eV on Fe-3d orbits for insulating FeBr_2 . The Hubbard U values are derived in Ref. [79] using the approach outlined in Ref. [80]. For CuMnAs , a metal, we used a smaller U value on Mn-3d orbits ($U=1.9$ eV, $J=0$ eV [40]) than for the insulator Ca_2MnO_4 ($U=3.9$ eV, $J=0$ eV)...”

In Figure 3 caption for the tetragonal CuMnAs we have added:

“The crystal and magnetic structure for tetragonal CuMnAs used in our DFT calculations are taken from Ref. [55].”

In Figure 4 caption for the tetragonal Ca_2MnO_4 we have added:

“The crystal and magnetic structure for tetragonal Ca_2MnO_4 used in our DFT calculations are taken from Ref. [48].”

In Figure 5 caption for the triagonal FeBr_2 we have added:

“The crystal and magnetic structure for triagonal FeBr_2 are taken from Ref. [65] and was tailored into a bilayer slab for the calculations with external electric field.”

Material specific parameters for other calculated materials in the supplementary information are provided in their corresponding Figure captions.

Referee Comment 4: Could they also plot the band structure along the complete conventional momentum space path and comment on the comparison of the obtained band structure with those reported in the literature? It is not clear what bands are plotted in Figure 3 (c), because in Figure 3 (b) several different bands cross the Brillouin zone.

Author Response: We have calculated the band structure along the complete conventional momentum space path. The results are presented in the supplementary information Figure S1. We have added a short comment comparing the current DFT results for CuMnAs to those reported in the literature in the supplementary information section B. We have also remade Figure 3, changing the less conventional k labelling to the conventional k labeling. The bands plotted in Fig. 3(c) are the highest two degenerate valence bands (denoted as V1 and V2). They are depicted by **red lines** in Figure S1.

“To compare with previously reported band structure results. In addition to Figure 3(c) in the main text, we calculated the band structure along the complete conventional momentum space path. The results are presented in Figure S1, which agrees well with those reported [SI ref. 1,2].”

Figure S1: Energy spectrum of antiferromagnetic tetragonal CuMnAs on the conventional k-paths. a with SOC turned off, and b with SOC turned on.

Referee Comment 5: The effect of spin-orbit interaction on the calculated energy bands and spin polarizations should be discussed.

a. The spin-orbit interaction cannot be turned off in real materials. How do the calculated projected spin polarizations given in the main text change when the spin-orbit interaction is included?

b. The authors argue that it is legitimate to use magnetic space groups with SOC to describe the spin polarisation of energy bands WITHOUT SOC. However, it is well-known that even energy bands differ when calculated with and without SOC. This is well established for paramagnetic materials and for magnetic materials it was discussed e.g. in Ref. 50 and Phys. Rev. X 12, 021016 (2022). Could the authors adapt this statement?

Author Response: For (a), the SOC-independent hidden spin polarization effect persists in the presence of SOC. This is because the effect being inherited from the unusual antiferromagnetic order rather than spin-orbit coupling. [PRB 102, 014422 (2020)]. Still, it is important to note the inclusion of SOC would modify the hidden spin polarization effect. We have added a paragraph discuss this in

Discussion section (page 10):

“...it is important to note the inclusion of SOC would modify the energy bands in both non-magnetic materials and magnetic materials [42, 66, 67]: (1) it reduces the degeneracy of certain bands which may cause additional spin splitting. (2) it mixes the spin polarized states of up and down (so spin is no longer a good quantum number), which results in momentum-dependent spin polarization that are not unidirectionally aligned; (3) it opens a gap for the crossing energy bands with opposite spin polarization. In compounds consist of low-Z elements the SOC-induced effect can be practically neglected.”

For (b), the referee is right that it is not legitimate to use magnetic space group with SOC to describe the spin degeneracy of the energy bands without SOC. We have adopted the referee’s statement (**page 10**):

“...the inclusion of SOC would modify the energy bands in both non-magnetic materials and magnetic materials [42, 66, 67] ...”

But we argued that it is legitimate to use magnetic symmetry with SOC to predict the occurrence or not of the SOC-independent spin splitting effect in magnetic materials. This is justified, as for magnetic materials there is a one-to-one correspondence between (a) the existence or absence of the UT in the spin space group and (b) the magnetic space group being type IV or type I/III. When it comes to filtering materials, we are not accessing the presence/absence of U (or UT) directly but the magnetic

space group type (i.e., type IV versus type I/III). We have edited this part of the text and added a paragraph to clarify this in the **Discussion section (pages 10,11)**:

“Use magnetic symmetry with SOC to describe the spin-splitting of energy bands without SOC: In collinear antiferromagnetic compounds, the existence of UT in the spin space group (symmetry group of the system without SOC) means there is a spatial translation T that connects the atomic sites with opposite magnetic moments and keeps the crystal structure invariant. By definition, antiferromagnets with primitive lattice translations that reverse the microscopic magnetic moments are known as having black and white Bravais lattice that is classified as magnetic space group (MSG) type-IV; Antiferromagnets without such translation T belongs to MSG type-I and type-III. [68] This suggests there is a one-to-one correspondence between (a) the existence or absence of the UT in the spin space group and (b) the magnetic space group being type IV or type I/III.

The correspondence relation can be formally established by introducing an auxiliary magnetic space group – a subgroup of the spin space group containing only elements of spatial and time reversal symmetries. This is referred to as “magnetic space group without SOC” in the Appendix of ref [39] or equivalently as “magnetic groups with pseudoscalar electron spin” in Ref [69]. Following that, we can prove a chain relation as depicted in Equation (1).

$$(a) \text{ UT} \leftrightarrow (b) \text{ } \Theta\text{T} \leftrightarrow (c) \text{ } \Theta\text{T} \leftrightarrow (d) \text{ MSG type IV} \quad (1)$$

spin space group Auxiliary MSG Standard MSG

(a) The existence or not of a UT symmetry in the spin space group corresponds to (b) the existence or not of a ΘT symmetry in the auxiliary group. This is because the ΘU symmetry preserves any collinear magnetic ordering and is a symmetry of any collinear antiferromagnets. [42-44]. Meanwhile, (b) the existence or not of a ΘT symmetry in the auxiliary magnetic space group (without SOC) corresponds to (c) the existence or not of a ΘT symmetry in the standard magnetic space group (with SOC). Antiferromagnetic materials whose (c) magnetic space group preserve (or violate) ΘT symmetry is classified as (d) MSG type-IV (or type-I/III). [68]

The established correspondence relation thus justifies the use of magnetic space group (with SOC) – avoiding the use of the “less familiar” spin symmetry [66] -- to predict whether the SOC-independent spin splitting effect will occur...”

Referee Comment 6: 4. The authors claim that effects observed despite enabling symmetry being absent can be explained by the “hidden effects” (e.g. lines 37-39, and 44-45). However, the authors do not prove the relationship between local projected spin polarization and observable global effects, e.g. due to an applied electric field. In fact, they show that induced spin splitting in FeBr₂ is proportional to electric field gating strength, which is not projection/sector dependent, but rather a global property. Could the authors find more compelling evidence of the relationship between the global property of spin splitting induced by a global electric field and the local spin polarization?

Author Response: To address this referee comment, we have remade Figure 5 to show the degenerate spin polarized states in FeBr₂ are localized on alternative individual layers/sectors, and edited the description of Figure 5 in the main text (**page 10**):

“...Because the applied electric field is small, the main characteristic of the observed spin polarization is inherited from the system without electric field. The layer-segregated states shown in Fig. 5b,c is thus a compelling evidence of the relationship between the global property of spin splitting induced by a global electric field and the local spin polarization...”

Figure 5: Revealing the hidden spin polarization in hexagonal FeBr₂ using an external electric field. **a** spin split band structure of FeBr₂ with a 10 meV/Å z-oriented external electric field. Red and blue lines represent the spin-up and spin-down polarized bands. Insert depicts the spin splitting between the bottom two conduction bands at Γ (denoted as Γ_{CB1} and Γ_{CB2}) as a function of the external electric field; **b** wavefunction plot for Γ_{CB1} ; and **c** wavefunction plot for Γ_{CB2} . The crystal and magnetic structure for trigonal FeBr₂ are taken from Ref. [65] and was tailored into a bilayer slab for the calculations with external electric field.

Referee Comment 7: 5. The authors should clarify their claims regarding the explanation of the observed spin polarization in NdBi and the manipulation of antiferromagnetism in CuMnAs.

a. In NdBi, Ref.51, spin polarization is reported in the surface spectrum/Fermi arcs, while the present authors focus on the bulk spectrum. How are the results of the bulk spectrum relevant to the surface spectrum?

b. CuMnAs: The quantization axes for local spin polarization and magnetic ordering are the same, and therefore it is not clear how the local spin polarization unrelated to spin-orbit coupling can torque the magnetic ordering itself. The claims in lines 200-202 about CuMnAs are thus not justified.

Author Response: For (a), we agree the spin polarization reported in the surface spectrum/Fermi arcs in Ref. 51 is mainly contributed from the low symmetry surface. We have now removed the discussion about Ref. 51 as suggested by the referee.

For (b): the SOC-independent hidden spin polarization in collinear antiferromagnet is parallel to the magnetic quantization axis, but the interplay with SOC could results in local spin polarization that is unparallel to the magnetic ordering quantization axes. We envision this may give rise to magnetic torques, but we don't know if the torque could rotate and switch the antiferromagnetic magnetic ordering simultaneously. Therefore, the suggested idea remains conceptual. Future studies that consider the coherent transport of the spin polarized electrons precess around the magnetic quantization axes when travelling through the atomic-thin sectors may be needed to proof this concept. For these sakes, we have removed the discussion parts on potential switching of the antiferromagnetic ordering as suggested by the referee and will discuss it in a future paper.

Referee Comment 8: 6. Some relevant references to previous work on local/global spin effects in antiferromagnets are omitted:

- a. How does the concept of "hidden sector" effects in this manuscript differ from the concepts of "hidden layer" effects discussed in Nature 595, 521 (2021) and references therein? It appears that the electric field-induced spin polarization in FeBr₂ (the same prototype as CuMnAs) is conceptually similar to the electric field-induced spin polarization in MnBi₂Te₄, see Figure 3g in Nature 595, 521 (2021).
- b. How do the present results compare to the local sector spin-polarisations calculated in CuMnAs previously, Phys. Rev. B 102, 125123 (2020).
- c. When citing "centrosymmetric antiferromagnets" (e.g. line 91), also some previous works on Kramers degeneracy in CuMnAs with combined space inversion and time-reversal symmetry should be cited: Phys. Rev. Lett. 118, 106402 (2017) and Phys. Rev. B 102, 125123 (2020)
- d. Spin splitting in antiferromagnets has also been reported in Phys. Rev. B 99, 184432 and Sci. Adv. 6 : eaaz8809 (unrelated to SOC), and in Phys. Rev. B 100, 245115 (related to SOC). The authors even use in Table in panel b of Figure 1 the BiCoO₃ and rutile crystals from these references as material examples.

Author Response: These references considering SOC to be a prerequisite in one or two specific materials are relevant but conceptually different from the current work where we performed systematic classification and exploration of the SOC independent hidden effects. We have cited these references in the introduction section and have briefly commented on them:

For (a): We note the concept of "hidden sector" is first introduced by some of our authors in 2014 in Nat. Phys. 10, 387-393 (2014) [6]. It is the same to that of the concept of "hidden layer" effect introduced in Nature 595, 521 (2021) [31]. But the mechanism is different. Ref [31] studies the layer Hall effect in the even-layered, two-dimensional MnBi₂Te₄ – an antiferromagnetic axion insulator, which requires SOC to couple the spin-up and spin-down bands. We have added in the **Introduction section (page 2):**

“(iv) X= “anomalous Hall effect” induced by SOC expected in ferromagnetic odd-layered MnBi₂Te₄ but observed in even-layered antiferromagnetic MnBi₂Te₄. [31]”

We have also added a comment about this paper on **page 9:**

“...We also note that the layer Hall effect in the even-layered MnBi₂Te₄ – in which electrons from the top and bottom layers spontaneously deflect in opposite directions but globally compensate – has been observed with the help of an applied electric field. [31]”

For (b), see our reply to comment 4 for comparison of the current results and the calculated results in Phys. Rev. B 102, 125123 (2020) [28]. We note that the corresponding author of [28] is also an author of the current manuscript. The paper studied the Dirac semimetal properties in a PT (time-reversal and inversion, here denoted as ΘI) preserved antiferromagnetic material CuMnAs which considers SOC to be a prerequisite.

For (c), we have cited Phys. Rev. Lett. 118, 106402 (2017) [46] and Phys. Rev. B 102, 125123 (2020) [28] as suggested on **page 4:**

“Antiferromagnets with ΘI symmetry [28,45,46] will not show such spin splitting.”

For (d), Phys. Rev. B 99, 184432 [34] and Sci. Adv. 6 : eaaz8809 [38] studied the apparent SOC-independent spin-splitting effect in antiferromagnets; Phys. Rev. B 100, 245115 [24] studied the apparent SOC-induced spin-splitting effect in antiferromagnetic BiCoO₃. We have cited these papers in the **Introduction section (page 2):**

“...here we discuss a different form of hidden effect whose corresponding apparent effect is independent of SOC [32-41].”

“...(iii) X= “spin polarization” induced by SOC was studied for antiferromagnetic systems...in non-centrosymmetric crystals (such as BiCoO3 [24])...”

Referee Comment 9: 7. The paper introduces some terminology, which, however, might be confusing for the general readership of Nature Communications:

a. The topic of the paper is the local spin polarisation in antiferromagnets and in principle only two prototypes are discussed - ferromagnetic (Figure 3/5) and antiferromagnetic (Figure 4). However, in Figure 1 the authors repeat the results/terminology from Phys. Rev. Materials 5, 014409 for global spin polarisation without citing it in the caption. Moreover, although the panel in Figure 1c is labelled as spin polarization sectors and refers to local spin polarisations of spin-degenerate antiferromagnets, they give examples of globally spin-polarized antiferromagnets/ferromagnets.

b. The authors refer to CuMnAs and Mn2Au as "centrosymmetric antiferromagnets", which can be misleading. If I understand correctly, the authors mean by this terminology that a CuMnAs/Mn2Au crystal in the paramagnetic state has an inversion, but not in the antiferromagnetic state (see also point 6 c.).

Author Response: For (a), all six prototypes are schematically illustrated in Figure 2 and exemplified by Figure 3,4,5 in the main manuscript and Figures S1-S5 in the supplementary information. Thus, all six prototypes are discussed, not only two. To improve the clarity, we have removed the confusing “global examples” for “spin-split sector” from Figure 1, we have cited Phys. Rev. Materials 5, 014409 [40] in figure 1 caption:

Figure 1: Hidden spin polarization in collinear antiferromagnets without SOC. a SOC-independent hidden spin polarization schematically illustrated as two copied of spin split energy bands localized on

sector- α and sector- β but globally mutually compensate; **b** three prototypes of spin degenerate bulk; **c** two prototypes of spin split sector. Sectors in **a** are represented by color-shaded planes, the red and blue lines in the plane represent the spin-up and spin-down bands. The spin-splitting prototypes in **b** defined for bulk [40] is generalized for sectors in **c**. Checkmark and cross in parentheses in **b** and **c** are used to indicate the presence or absence of the symmetry.

Fig. 1b,c is not a repetition of the results/terminology from our earlier publications PRB 102, 014422 (2020) [39], Phys. Rev. Materials 5, 014409 (2021) [40]. It introduces the necessary terminology of “spin-splitting prototypes (SST)” for bulk and generalizes the concept to sectors. We have added a clarification on **page 4**:

“The classification defined in bulk crystals [39,40] can be generalized to sectors of a bulk based on the local sector symmetry. Fig. 1b,c. summarizes the classification of “spin degenerate bulk” vs “spin-split sector”. This will later be applied to describe the symmetry conditions and to define the different prototypes for the hidden spin polarization effect in antiferromagnets.”

For (b), we have added in the **introduction section (page 2)** a clarification on what we mean by centrosymmetric:

“(iii) X= “spin polarization” induced by SOC was studied for antiferromagnetic systems. The effect is again expected only in non-centrosymmetric crystals (such as BiCoO₃ [24]) but shown in centrosymmetric crystals (such as CuMnAs and Mn₂Au [25-28] where combined symmetry of inversion and time reversal disallows splitting. Here, “centrosymmetric” means the crystal in the non-magnetic state has an inversion.”

Referee Comment 10: 8. Some of the typos should be corrected, such as “Dresslhouse” (line 50), or missing characters in the caption of supplementary Figure S5 (lines 114-115).

Author Response: We have corrected these typos.

Reviewer #2 comments and author response

Referee Comment 1: In the manuscript under consideration, the authors use a “top-down” strategy to engineer antiferromagnetic materials featuring a hidden spin-polarization. The idea works in the following way: a careful analysis of the underlying symmetries suggests that sectors characterized by apparent (i.e. visible) spin-polarization can be stacked together, in a compensated way, in order to feature a hidden spin-polarization in the bulk. By hidden spin-polarization, the authors meant a finite spin expectation value when the Bloch wave functions are projected onto the individual sectors (layers). What I find interesting in this work is that the spin-polarization is achieved without the requirement of spin-orbit coupling (SOC). Indeed, the spin-polarization of each sector is achieved either by AFM or FM ordering. Driven by the symmetry analysis, the authors then scrutinized a vast database of known collinear AFM materials, and performed first-principles calculations on several candidates compounds to bring evidence of the solidity of their “top-down” strategy.

This work represents an important step further beyond the already known hidden Rashba and Dresselhaus spin-splittings that unavoidably require a sizable contribution from SOC. Here, on the other hand, the concept of “hidden” is extended to low-Z compounds, for which SOC is most of the

times a tiny perturbation. I find the proposal convincing and potentially relevant to further boost the theoretical and experimental efforts in searching for new spin-polarized materials. I thus recommend its publication in Nature Communications.

Author Response: To further highlight the importance of this work, we have slightly edited the **Abstract (page 1):**

“...Here, we discuss hidden spin polarization effect in collinear antiferromagnets without the requirement for spin-orbit coupling (SOC). Symmetry analysis suggests that antiferromagnets hosting such effect can be classified into six types depending on the global vs local symmetry. We identify which of the possible collinear antiferromagnetic compounds will harbor such hidden polarization and validate these symmetry enabling predictions with first-principles density functional calculations for several representative compounds. This will boost the theoretical and experimental efforts in finding new spin-polarized materials.”

and **Introduction section, last paragraph (page 2,3)**

“...Here we discuss a different form of hidden spin polarization effect (see Fig. 1(a)) whose corresponding apparent effect is independent of SOC [32-41]; And the hidden effect exists in antiferromagnetic materials where spin-up and spin-down bands are paired. This represents a step further beyond the already known hidden Rashba and hidden Dresselhaus spin-polarization that unavoidably require a sizable contribution from SOC. A careful analysis of the “global (bulk) vs local (sector)” symmetries suggests that antiferromagnets hosting the SOC-independent “hidden” spin polarization effect can be delineated into six types. We scrutinize a vast database of known collinear AFM materials and performed first-principles calculations on several selected candidate compounds assuming zero SOC. We show that such hidden, SOC-independent effects reflect the intrinsic properties of the perfect crystal rather than an effect due to imperfections. The interest in this SOC-independent hidden spin polarization effect stems both from the evolving of the fundamental understanding of general hidden effects in solids, and from the ability to extend the pool of useful materials for potential spintronic applications.”

Reviewer #3 comments and author response

Referee Comment 1: The manuscript by **Lin-Ding Yuan, Xiuwen Zhang, Carlos Mera, and Alex Zunger** is devoted to an interesting feature of electron energy bands in collinear antiferromagnets with negligible spin-orbit interaction, namely to the spin polarization of the bands which can be revealed only after a projection on different sectors of the systems is carried out. The authors call this phenomenon a 'hidden' spin polarization and put it in a broader context of effects that occur in a system due to its symmetry. Based on the symmetry arguments, the authors define six different types of the hidden spin polarization, identify potential candidate materials exhibiting the effect by filtering a database of magnetic materials, and discuss the response of the electron bands to external electric fields as well as the potential applicability. The authors' statements are illustrated by figures presenting results of electronic-structure calculations for selected compounds, obtained by up-to-date techniques.

On one hand, the essence of the phenomenon is nearly trivial, as can easily be found by inspecting the electron bands of the simplest one-dimensional model of an antiferromagnet (with two sites in the unit cell, described with a single orbital per site and a spin-independent nearest-neighbor hopping). On the other hand, however, the systematic classification of various antiferromagnets and their sectors (ferro- or antiferromagnetic), and the outlined relevance for the whole area of spintronics will undoubtedly attract attention both by theoreticians and by experimentalists working in the field. It should also be

noted that collinear antiferromagnets with weak spin-orbit interaction are intensively studied at present, as can be documented, e.g., by a very recent review Smejkal et al., Phys. Rev. X 12, 040501 (2022). For these reasons, I believe that the authors' manuscript deserves publication in a reputable scientific journal.

Author Response: We thank the referee for the comment. The hidden spin polarization in antiferromagnets discussed in this work is not trivial. In this work, we present a systematic classification of antiferromagnets hosting the SOC-independent hidden spin polarization. The simple one-dimensional model (mentioned by the referee), considers an antiferromagnet as two copies of ferromagnetic chain that intercalate, only represents the most trivial one of the six AFM types we discussed in this work (bulk belonging to SST-2, sector belonging to SST-5). Other types, especially the three types where hidden spin polarization arises from individual antiferromagnetic sector, requires the constitute sector to preserve or violate certain symmetries are exotic.

Referee: In my view, however, the work can be improved in two directions:

Referee Comment 2: 1) Proper characterization of the underlying symmetry groups.

The authors use the standard magnetic space groups (Shubnikov space groups, relevant for electron systems with spin-orbit interaction) for a treatment of systems without spin-orbit interaction. This seems advantageous over approaches based on 'less familiar' and more exotic spin groups featured by two independent rotations (in the configuration space and in the spin space). However, their formalism employs an operation U (a spin rotation acting only in the spin space that causes a spin reversal) which does not belong among the basic operations of the traditional magnetic groups. It seems to me that this addition of U to the other symmetry operations moves the actual group formalism towards the spin groups. Readers would like to know whether the complete group (i.e., the magnetic space group extended by the spin-only operation U) is equivalent to the full spin group, or to its subgroup, or whether the authors introduce implicitly a new kind of groups.

On a more pragmatic level, one would like to know how the filtering using the MAGNDATA information (with the symmetry of each system defined by the magnetic space group) was carried out. I think that the presence/absence of the spatial translations (T), time reversal (θ), the spatial inversion (I), and their combinations can be obtained in a straightforward manner (in principle, just by looking at the list of elements of the magnetic space group). However, the way to assess the presence/absence of U (or UT) among the symmetry elements of the system only from its magnetic space group alone is unclear to me.

Author Response: We thank the referee for prompting us to clarify our claims on the symmetry. Indeed, the operation U is a pure spin rotation which does not belong among the basic operations of the traditional magnetic groups. As such, the UT symmetry belongs to the spin space group of the material without SOC. This provides the formal basis for our symmetry analyze. The legitimate of using standard magnetic space group with spin-orbit interaction for a treatment of magnetic systems without spin-orbit interaction is based on the one-to-one correspondence between (a) the existence or absence of the UT in the spin space group (symmetry group of the system without SOC) and (b) the magnetic space group (symmetry group of the system with SOC) being type IV or type I/III. When it comes to filtering materials, we are not accessing the presence/absence of U (or UT) directly but the magnetic space group type (i.e., type IV versus type I/III). MSG type-II that support nonmagnetic

materials are excluded from our analysis. To justify our claims on symmetry, we have edited the related text and added a paragraph in the Discussion section:

“...The correspondence relation can be formally established by introducing an auxiliary magnetic space group – a subgroup of the spin space group containing only elements of spatial and time reversal symmetries. This is referred to as “magnetic space group without SOC” in the Appendix of ref [39] or equivalently as “magnetic groups with pseudoscalar electron spin” in ref [69]. Following that, we can prove a chain relation as depicted in Equation (1).

$$\begin{array}{ccccccc} \text{(a) UT} & & \text{(b) } \Theta T & & \text{(c) } \Theta T & & \text{(d) MSG type IV} \\ \text{spin space group} & \leftrightarrow & \text{Auxiliary MSG} & \leftrightarrow & \text{Standard MSG} & \leftrightarrow & \end{array} \quad (1)$$

(a) The existence or not of a UT symmetry in the spin space group corresponds to (b) the existence or not of a ΘT symmetry in the auxiliary group. This is because the ΘU symmetry preserves any collinear magnetic ordering and is a symmetry of any collinear antiferromagnets. [42-44]. Meanwhile, (b) the existence or not of a ΘT symmetry in the auxiliary magnetic space group (without SOC) corresponds to (c) the existence or not of a ΘT symmetry in the standard magnetic space group (with SOC). Antiferromagnetic materials whose (c) magnetic space group preserve (or violate) ΘT symmetry is classified as (d) MSG type-IV (or type-I/III). [68] ...”

Referee Comment 3: 2) Relation of the introduced classification to the response behavior.

The various kinds of the response of a bulk system to an external electric field lead to well-known quantities, such as the charge and spin conductivity, but also to novel quantities, such as the current-induced spin torques or current-induced spin polarization. I wonder whether the authors can establish a relation between the six different types of the collinear antiferromagnets and the vanishing/nonvanishing values of a transport response quantity. This could serve as a guide in search for systems exhibiting that particular response behavior and it would underline the importance of the introduced classification scheme.

Author Response: We agree that the response behavior of transport properties in the collinear antiferromagnets with hidden spin polarization is an important consideration, however, establish a relation between the hidden spin polarization in collinear antiferromagnets and the vanishing/nonvanishing values of a transport response quantity is beyond the scope of the current study. We now added a brief comment on this point under “electric field control” in the discussion section on page 11:

“In fact, the electric field applied couples with the electron spin through the magnetoelectric effect [70], which is only allowed under specific symmetry conditions. [45] Additionally, transport properties that are even functions of the sectors can take non-vanishing values in a hidden system. For example, non-reciprocal nonlinear current respond to an applied electric field is recently demonstrated in antiferromagnetic tetragonal CuMnAs. [71] This serves as a guide in search for other prototypic systems exhibiting this particular response behavior.”

45. Zhao, H. J. et al. Zeeman Effect in Centrosymmetric Antiferromagnetic Semiconductors Controlled by an Electric Field. *Physical Review Letters* 129, 187602

70. Fiebig, M. Revival of the magnetoelectric effect. *Journal of Physics D: Applied Physics* 38, R123

71. Chen, W., Gu, M., Li, J., Wang, P. & Liu, Q. Role of Hidden Spin Polarization in Nonreciprocal Transport of Antiferromagnets. *Physical Review Letters* 129, 276601

Referee Comment 4: I have also found two minor points to be corrected: 3) In the caption to Figs. 3 and 4 (and also to Figs. S1 - S4 of the Supplement), the expression 'polynomials' should perhaps be replaced by 'polyhedra'.

Author Response: The referee is right. We have corrected these typos.

Referee Comment 5: 4) In the text of the Supplement, in the parts describing the results for La₂NiO₄ and MnS₂ systems, one can read 'The two ferromagnetically ordered Ru₂O₇ sectors are ...', which should be corrected.

Author Response: We have corrected these typos.

Referee Comment 6: In summary, I find the submitted work interesting and potentially important for the field of spintronics. Nevertheless, the work can be improved along the indicated directions, which could strengthen its impact on the physical community.

Author Response: Thank you.

Reviewers' Comments:

Reviewer #1:

None

Reviewer #3:

Remarks to the Author:

In the revised version of their work, the authors addressed properly all particular concerns mentioned in my previous report. This improved the level of presentation of the manuscript to my full satisfaction.

I have found only one minor point (misprint) in the revised version:

In the inserted Eq. (1), 'Starndard MSG' contains an extra 'r'.

A point-by-point response to the referees' comments:

Reviewer #3 comments and author response

Referee Comment 1: In the revised version of their work, the authors addressed properly all particular concerns mentioned in my previous report. This improved the level of presentation of the manuscript to my full satisfaction.

Author reply 1: Thank you

Referee Comment 2: I have found only one minor point (misprint) in the revised version: In the inserted Eq. (1), 'Starndard MSG' contains an extra 'r'.

Author reply 2: We have corrected this typo in the main text.